# TSD-Net: A Traffic Sign Detection Network Addressing Insufficient Perception Resolution and Complex Background

**DOI:** 10.3390/s25113511

**Published:** 2025-06-02

**Authors:** Chengcheng Ma, Chang Liu, Litao Deng, Pengfei Xu

**Affiliations:** School of Artificial Intelligence, Beijing Normal University, Beijing 100875, China; machengcheng@mail.bnu.edu.cn (C.M.); lc826@mail.bnu.edu.cn (C.L.); litao@mail.bnu.edu.cn (L.D.)

**Keywords:** traffic sign detection, small object detection, complex background, feature enhancement module, adaptive feature fusion, dynamic convolution, intelligent transportation systems

## Abstract

With the rapid development of intelligent transportation systems, traffic sign detection plays a crucial role in ensuring driving safety and preventing accidents. However, detecting small traffic signs in complex road environments remains a significant challenge due to issues such as low resolution, dense distribution, and visually similar background interference. Existing methods face limitations including high computational cost, inconsistent feature alignment, and insufficient resolution in detection heads. To address these challenges, we propose the Traffic Sign Detection Network (TSD-Net), an improved framework designed to enhance the detection performance of small traffic signs in complex backgrounds. TSD-Net integrates a Feature Enhancement Module (FEM) to expand the network’s receptive field and enhance its capability to capture target features. Additionally, we introduce a high-resolution detection branch and an Adaptive Dynamic Feature Fusion (ADFF) detection head to optimize cross-scale feature fusion and preserve critical details of small objects. By incorporating the C3k2 module and dynamic convolution into the network, the framework achieves enhanced feature extraction flexibility while maintaining high computational efficiency. Extensive experiments on the TT100K benchmark dataset demonstrate that TSD-Net outperforms most existing methods in small object detection and complex background handling, achieving 91.4 mAP and 49.7 FPS on 640 × 640 low-resolution images, meeting the requirements of practical applications.

## 1. Introduction

With the rapid development of intelligent transportation systems, autonomous driving technology has imposed stricter requirements for the accuracy and real-time performance of environmental perception. Among these systems, traffic signs, as key visual information carriers in road traffic systems, carry complex semantic information. Therefore, traffic sign detection is one of the most important applications of object detection in the field of intelligent transportation. Efficient and accurate automated detection of traffic signs helps ensure safe driving and prevents frequent traffic accidents [1].

Given the important role of traffic sign detection in intelligent transportation systems, developing efficient and robust detection algorithms has become an important research topic in this field [2,3]. With the rapid development of deep learning technology, deep learning-based object detection algorithms, such as Faster R-CNN [4], SSD [5], and YOLO [6], have become the mainstream choice for optimizing traffic sign detection problems due to their powerful feature extraction capabilities. However, despite the success of these object detectors for general purposes, traffic sign detection still faces unique challenges that make mainstream detection methods insufficient for practical applications. Due to the resolution limitations of autonomous driving cameras, distant traffic signs often appear as low-resolution features in the captured images. Taking the TT100K dataset [7] as an example, despite the image resolution of up to 2048 × 2048, more than 40% of traffic signs occupy an area of less than 32 × 32 pixels. After passing through multiple pooling layers, the features of these small-sized objects suffer from significant information loss [8], which may affect the detection performance.

When multiple small traffic signs are closely positioned, the output detection head of conventional detectors lacks perception resolution to effectively distinguish adjacent signs, leading to a relatively severe issue of missed detections in traditional object detection methods, as shown in Figure 1a. Additionally, in road environments, complex backgrounds remain a significant challenge for object detection. Urban environments often contain a large number of visually similar interfering elements, such as billboards and store signs. These elements exhibit strong similarities with traffic signs in terms of shape and color, thereby increasing the risk of missed detections and false positives [9], as illustrated in Figure 1b.

To address the insufficient resolution in small object detection, recent research has introduced contextual background information [10,11,12]. However, these methods often face challenges when processing densely distributed small objects, as background noise may obscure the target’s intrinsic features [13]. Some attention mechanism-based improvement methods have enhanced target features [14,15,16], but they incur substantial computational overhead when dealing with densely distributed small objects, and none of these methods resolve the fundamental resolution issue. While multi-scale feature fusion methods such as feature pyramids [17,18,19] can improve small object detection performance, the feature misalignment between shallow semantic features and deep semantic features [20] negatively impacts long-distance traffic sign detection.

For complex background detection optimization, recent methods predominantly employ Transformer-based approaches with multi-head self-attention mechanisms [21,22]. However, when using high-resolution feature maps, computational costs increase quadratically [23]. When processing small objects and partially occluded traffic signs, global attention mechanisms may be dominated by significant features, resulting in the loss of local detail information [24]. Although two-stage detection methods proposed by Li et al. [8] have achieved good detection results, they introduce additional computational overhead and longer inference times.

To address this, we propose the Traffic Sign Detection Network (TSD-Net), an improved detection framework with the following targeted improvements. To enhance the model’s expressiveness, we innovatively combine the C3k2 module with dynamic convolution [25], which adaptively generates convolution kernel weights to increase flexibility in feature extraction while reducing FLOPS. To handle the complex urban environments, we incorporate a feature enhancement module [26] to strengthen the model’s ability to capture target details while suppressing background interference. To address the insufficient resolution problem in small traffic sign detection, we enhance the feature pyramid architecture with a refined high-resolution detection branch and introduce an adaptive spatial feature fusion [27] detection head with dynamic convolution [25] for optimal cross-scale feature fusion, which effectively preserves fine-grained features of small objects. Through these improvements, our method effectively enhances the detection performance for traffic signs, especially in challenging scenarios with small objects and complex backgrounds.

In summary, the main contributions of this paper are as follows:(1)We propose TSD-Net, an improved traffic sign detection framework that innovatively incorporates a dynamic convolution mechanism [25] into the C3k2 module of the network, enhancing the flexibility of feature extraction through adaptively generated convolution kernel weights.(2)We integrate a Feature Enhancement Module (FEM) [26], improving the model’s capability to capture object details while maintaining robustness against complex backgrounds.(3)We enhance the feature pyramid structure by introducing a high-resolution detection head and designing an Adaptive Dynamic Feature Fusion (ADFF) detection head, combining spatial feature fusion [27] with dynamic convolution [25] to achieve optimal cross-scale feature fusion, significantly improving the detection capability for small traffic signs.(4)Experiments on the TT100K [7] benchmark dataset demonstrate that our method achieves state-of-the-art performance in small object detection and complex background processing while maintaining real-time processing efficiency suitable for practical application scenarios.

## 2. Related Works

### 2.1. Low-Resolution Object Detector

Traffic sign detection represents a critical challenge in autonomous driving, particularly concerning small and low-resolution object detection. Existing approaches can be broadly categorized into traditional techniques and deep learning methods, with the latter further divided into single-stage and two-stage detection algorithms. Single-stage detection algorithms such as YOLO, SSD [5], and RetinaNet [28] have been widely adopted in autonomous driving scenarios due to their real-time performance capabilities. YOLO formulates detection as an end-to-end regression problem; SSD enhances small object detection accuracy through multi-layer feature detection; RetinaNet introduces Focal Loss to address class imbalance issues.

The investigation into low-resolution object detection by Wang et al. [29] revealed that substantial variations in the dimensions of traffic signs resulted in fluctuations in the accuracy of detection. Furthermore, it was observed that conventional feature pyramid networks exhibited deficiencies in the extraction of multi-scale features. To overcome these limitations, they proposed an enhanced feature pyramid model incorporating adaptive attention modules and feature enhancement components, thereby reducing information loss during multi-scale feature fusion. In related research, Chen et al. [30] introduced a multi-scale channel attention mechanism to enhance low-level texture feature recognition. In another study, Chen et al. [31] designed a U-Net-like architecture that extracts multi-level features while compressing channels and implements feature compensation during upsampling, effectively reducing feature information loss.

Two-stage detection algorithms such as the R-CNN series demonstrate improved detection accuracy but typically fail to meet real-time processing requirements. For instance, Liang [32] proposed an enhanced Sparse R-CNN that integrates coordinate attention modules with ResNeSt, enabling extracted features to focus more effectively on salient information. Li et al. [8] developed a two-stage fusion network that directly predicts categories in the initial detection phase and merges these predictions with second-stage results while also implementing Surrounding-Aware Non-Maximum Suppression (SA-NMS) technology to filter detection boxes more efficiently. Nevertheless, when processing small objects, these two-stage approaches may generate suboptimal candidate boxes from the region proposal generator, resulting in diminished detection accuracy for small objects.

### 2.2. Complex Background Object Detector

Complex background object detection represents a significant research challenge in the field of object recognition. Li et al. [33], in their comprehensive review, identified that complex background environments primarily consist of adverse weather conditions (such as fog, heavy rain, and snow), object occlusions (such as trees and buildings), and complex sea conditions (such as sea fog and strong waves). These factors result in low signal-to-noise ratios and insufficient contrast between targets and backgrounds, severely impacting detection performance. Under complex background interference, target features are easily obscured, leading to inadequate feature representation, target distortion, and inaccurate localization issues.

Researchers have proposed various solutions to address complex background interference. Noh et al. [34] introduced a feature-level super-resolution direct supervision method that effectively mitigates the interference of complex backgrounds on target features by enhancing feature quality. Wu et al. [35] developed a cascade mask generation framework that efficiently integrates multi-scale feature information, improving system robustness in complex scenarios. Wei et al. [36] proposed RFAG-YOLO, which enhances the model’s capability to recognize targets in complex backgrounds through novel feature fusion processes and attention mechanisms, demonstrating exceptional performance particularly in small target detection within UAV imagery. Addressing the issue of insufficient feature representation in complex backgrounds, Zhou et al. [37] incorporated context-aware information into the detection framework, capturing environmental information at different levels through multi-scale feature fusion to enhance the model’s discriminative capability for targets in complex backgrounds. However, these methods exhibit certain limitations in integrating local–global information, resulting in difficulties in effectively distinguishing morphologically similar targets and backgrounds in complex road environments.

## 3. Materials and Methods

### 3.1. Network Architecture of TSD-Net

Figure 2 illustrates the overall architecture of our proposed TSD-Net. To address the aforementioned challenges of low resolution and complex backgrounds, our TSD-Net focuses on two key approaches: effective feature extraction fusion and receptive field expansion. We built the network architecture based on YOLO11 [38]. To enhance the model’s expressiveness and lightweight performance, we upgraded the C3k2 module in the network to a C3k2-Dynamic module, improving the model’s feature representation capability by increasing parameter count while avoiding significant increases in FLOPs. To tackle the challenge of complex background interference, we integrated a Feature Enhancement Module (FEM) in the neck section, which enhances the model’s robustness against complex backgrounds by expanding the receptive field. To address the insufficient resolution issue in small traffic sign detection, we redesigned the feature pyramid network and constructed a high-resolution detection head to effectively preserve target details. Additionally, we designed the ADFF detection head, which combines feature map weights generated from ASFF [27] and dynamic convolution [25], achieving efficient integration of low-level and high-level features and significantly improving the network’s capability to detect densely distributed small objects.

Compared to recent works, while YOLOv8 [39] has established a solid foundation for general object detection, its inherent limitations in feature representation capability restrict its precision in detecting small traffic signs. To address this issue, our C3k2-Dynamic module implements an adaptive feature extraction mechanism that enhances representational capacity while maintaining computational efficiency. Unlike TSD-YOLO [14], which focuses on long-range attention mechanisms and dual-branch architecture, our Feature Enhancement Module (FEM) tackles complex background interference through expanded receptive fields. Furthermore, while Transformer-based architectures typically incur substantial computational overhead and suffer from feature misalignment issues [40], our Adaptive Dynamic Feature Fusion (ADFF) detection head achieves efficient multi-scale feature integration through a computationally efficient architecture.

In summary, our approach addresses the limitations of previous traffic sign detection studies in low-resolution and complex background scenarios through the synergistic integration of three complementary modules. First, the C3k2-Dynamic module establishes a rich feature representation foundation while maintaining computational efficiency. These features are enhanced through the FEM, which leverages receptive field expansion to effectively distinguish traffic signs in complex visual environments. Subsequently, the ADFF detection head coordinates adaptive multi-scale feature fusion to preserve fine-grained spatial details. Figure 3 shows our component block diagram.

### 3.2. C3k2-Dynamic Module

In traffic sign detection tasks, feature extraction capability directly impacts detection performance. Traditional convolutional neural networks process all input features with fixed-weight convolutional kernels, lacking adaptability to different feature contents, which limits feature representation capability [41]. To address this issue, we propose the C3k2-Dynamic module, which is based on the CSP structure and incorporates dynamic convolution mechanisms to achieve adaptive processing of different input features.

The overall architecture of the C3k2-Dynamic module is shown in Figure 4a. Consistent with the C3k2 module, when the C3k parameter is True, C3k2-Dynamic will use the C3k module; otherwise, it will use the Bottleneck module. The C3k2-Dynamic module retains the basic CSP structure of C3k2 but replaces the standard Bottleneck with a dynamic convolution Bottleneck, as illustrated in Figure 4b. The dynamic convolution Bottleneck consists of two key components: the first part is a standard 1×1 convolution for channel dimensionality reduction to decrease subsequent computational costs; the second part is a dynamic convolution layer which, unlike the fixed convolutional kernels in standard convolution, can adaptively adjust convolutional weights based on input features, as shown in Figure 5.

There are significant structural differences between conventional convolution and dynamic convolution. Conventional convolution processes input features *X* using only a single fixed kernel, while dynamic convolution fuses parameters from multiple expert kernels to create a richer parameter space. For input features *X*, our dynamic convolution layer with *M* experts (M=4) is defined as [25]:(1)$$\mathrm{DynamicConv}\left(X\right)=X*W^{\prime },\;W^{\prime }=\sum_{i=1}^{M}\pi_{i}\left(X\right)\cdot W_{i}$$
where ∗ is the convolution operation, $W_{i}$ represents the weight tensor of the *i*-th expert convolution (kernel size K=2), and $\pi_{i}\left(X\right)$ is the dynamically generated weight coefficient. These coefficients are obtained through a routing function:(2)$$\pi_{i}\left(X\right)=\sigma \left(V_{i}\cdot \mathrm{Pool}\left(X\right)+c_{i}\right)$$
where Pool(X) represents the global average pooling layer, $V_{i}$ and $c_{i}$ are learnable parameters, and σ denotes the sigmoid function that ensures coefficient values range between 0 and 1. Additionally, our choices regarding the number of expert kernels (*M*), kernel size (*K*), and the implementation of sigmoid function in the routing mechanism will be validated through comprehensive ablation studies presented in Section 4.2.1. Although multiple expert kernels are introduced, the computation does not involve *M* separate convolution operations. Instead, our approach first generates fusion weights through the routing function then linearly combines *M* expert kernels into a single effective convolution kernel and finally performs only one standard convolution operation. Consequently, the computational overhead introduced by parameter generation and weight fusion steps is negligible compared to the convolution operation itself [25].

After improvement, the dynamic convolution mechanism enables the model to adaptively adjust convolution strategies based on input feature content, demonstrating significant advantages over traditional fixed-kernel convolution. In traffic sign detection tasks within road environments, dynamic convolution leverages adaptive parameters to accommodate highly variable contexts while effectively mitigating class imbalance issues in datasets by providing more flexible feature extraction for rare sign categories. Furthermore, its multi-expert architecture substantially expands the model’s parameter space without proportionally increasing computational complexity, making it more suitable for deployment in autonomous driving systems.

### 3.3. Feature Enhancement Module (FEM)

Due to the complexity of road environments, small-sized traffic sign detection tasks frequently encounter traffic signs with similar features. However, traditional YOLO architectures exhibit limited extraction capabilities. Features extracted at this stage contain minimal semantic information and narrow receptive fields, making it challenging to differentiate between small-sized traffic signs and background elements [12,16]. To address this issue, we introduce the Feature Enhancement Module (FEM) [26] to enhance the backbone network’s ability to extract features from small-sized traffic signs. FEM effectively enhances small object feature representation and distinguishes background through a three-fold mechanism: first, it employs a multi-branch convolutional structure to extract multi-dimensional discriminative semantic information, enabling the model to simultaneously attend to various features of traffic signs, including shape, color, and texture [42]; second, it utilizes atrous convolution to obtain more abundant local contextual information, effectively expanding the receptive field range. When backgrounds are complex, a larger receptive field helps the model understand the relationship between targets and their surroundings [43]. Finally, through a residual structure, it forms an equivalent mapping feature map to preserve critical feature information of small objects, ensuring that original discriminative features are not lost while enhancing feature representation [44].

The overall architecture of FEM is illustrated in Figure 6. This module comprises two branches equipped with atrous convolution. Each branch initially performs a 1 × 1 convolution operation on the input feature map to optimize channel dimensions for subsequent processing. The first branch is constructed as a residual structure, forming an equivalent mapping feature map to preserve critical feature information of small objects. The remaining three branches perform cascaded standard convolution operations with kernel sizes of 1 × 3, 3 × 1, and 3 × 3, respectively. Notably, the middle two branches introduce atrous convolution, enabling the extracted feature maps to retain richer contextual information. Finally, the outputs from all branches are concatenated along the channel dimension, generating a comprehensive feature representation containing multi-scale and multi-directional information.

The mathematical representation of the FEM can be formalized as follows:(3)$$W_{1}=f_{3\times 3\_conv}\left(f_{1\times 1\_conv}\left(F\right)\right)$$(4)$$W_{2}=f_{3\times 3\_diconv}\left(f_{3\times 1\_conv}\left(f_{1\times 3\_conv}\left(f_{1\times 1\_conv}\left(F\right)\right)\right)\right)$$(5)$$W_{3}=f_{3\times 3\_diconv}\left(f_{1\times 3\_conv}\left(f_{3\times 1\_conv}\left(f_{1\times 1\_conv}\left(F\right)\right)\right)\right)$$(6)$$Y=Cat\left(W_{1},W_{2},W_{3}\right)⊕f_{1\times 1\_conv}\left(F\right)$$
where $f_{1\times 1\_conv}$, $f_{1\times 3\_conv}$, $f_{3\times 1\_conv}$, and $f_{3\times 3\_conv}$ denote standard convolution operations with kernel sizes of 1 × 1, 1 × 3, 3 × 1, and 3 × 3, respectively; $f_{3\times 3\_diconv}$ represents atrous convolution with a dilation rate of 5 (the optimal value determined through ablation studies, as detailed in Section 4.2.2); Cat(·) indicates the concatenation operation of feature maps along the channel dimension; ⊕ denotes element-wise addition of feature maps; *F* is the input feature map; $W_{1}$, $W_{2}$, and $W_{3}$ represent the output feature maps of the three branches after convolution operations; and *Y* is the final output feature map of the FEM.

Through this design, FEM effectively addresses the insufficient feature representation problem in small object detection. The multi-branch structure enables the module to extract discriminative features from different dimensions while the atrous convolution increases the receptive field, which is conducive to learning richer local contextual features and enhancing the ability to capture contextual information.

### 3.4. Adaptive Dynamic Feature Fusion (ADFF) Detection Head

The Feature Pyramid Network (FPN) [45] enhances the hierarchical feature representation of CNNs through a top-down pathway, significantly improving multi-scale object detection performance. YOLO11 [38] adopts an enhanced FPN structure, achieving efficient multi-scale object detection through multi-level feature fusion. This structure contains bottom-up and top-down feature transmission paths and introduces a PSA attention mechanism to enhance feature representation capability. YOLO11’s detection heads are distributed across three feature layers with different resolutions, specifically at 1/8, 1/16, and 1/32 of the input image resolution, corresponding to small-, medium-, and large-scale object detection. However, for most small-sized traffic signs, this FPN architecture still lacks ideal resolution [46]. Furthermore, YOLO11 primarily employs simple element-wise addition or feature concatenation during feature fusion; these static fusion methods struggle to adaptively adjust fusion strategies according to variations in different scenes and object sizes [47]. Although feature maps at different scales contain complementary semantic and spatial information, how to effectively integrate this information to improve detection performance remains a challenge to be addressed.

To solve the aforementioned problems, we redesigned the FPN structure. First, by adding a small object detection layer in the 4× downsampled high-resolution feature map, we enable the backbone network to extract multi-scale features {P2,P3,P4,P5}, where the highest-resolution P2 layer contains rich detail information particularly suitable for small-sized traffic sign detection. Furthermore, to address the feature misalignment issue between shallow and deep layers, we propose an Adaptive Dynamic Feature Fusion (ADFF) detection head inspired by ASFF [27]. As shown in Figure 7, ADFF includes two key steps: feature rescaling and adaptive dynamic fusion. The feature rescaling step precisely aligns features from different scales while preserving their unique characteristics. The adaptive dynamic fusion step then determines the optimal contribution of each scale at each spatial location. Unlike existing methods [27], ADFF implements a dual-layer adaptive mechanism with dynamic convolution in the weight generation network, allowing the model to dynamically adjust fusion weights based on input content.

Feature Rescaling. In our improvement of FPN, we denote the feature maps from different levels of YOLO11 as $x_{l}$, where l∈{0,1,2,3} corresponds to the four feature levels P2,P3,P4, and P5, respectively. To achieve cross-scale feature fusion, we need to adjust all feature maps to the same spatial dimensions. For instance, for the first layer, we rescale feature maps $x_{n}$ from other levels *n*(n≠l) to match the shape of $x_{l}$. Given that the four levels of YOLO11 have varying resolutions and channel dimensions, we employ different strategies for upsampling and downsampling operations. For upsampling operations, we first use a 1×1 convolutional layer to compress the feature map’s channel dimensions to match the target level, followed by nearest-neighbor interpolation to enlarge the feature map resolution to the target size. For example, when the target level is P3, the P5 feature map requires 4× upsampling, while P4 needs 2× upsampling. For downsampling operations, we implement a multi-level downsampling strategy. When the downsampling ratio is 1/2, we utilize a 3×3 convolutional layer with a stride of 2 to simultaneously adjust channel dimensions and reduce resolution; when the downsampling ratio is 1/4, we first apply a max-pooling layer with a stride of 2, followed by a convolutional layer with a stride of 2; for larger downsampling ratios (such as 1/8), we achieve this through cascading multiple 2× downsampling operations.

Adaptive Dynamic Fusion. For a feature vector at position (i,j) in each feature map, the fusion process can be represented as:(7)$$y_{ij}^{l}=\alpha_{ij}^{l}\cdot x_{ij}^{0\to l}+\beta_{ij}^{l}\cdot x_{ij}^{1\to l}+\gamma_{ij}^{l}\cdot x_{ij}^{2\to l}+\delta_{ij}^{l}\cdot x_{ij}^{3\to l}$$
where $y_{ij}^{l}$ represents the feature vector of the output feature map $y^{l}$ at position (i,j), and $\alpha_{ij}^{l}$, $\beta_{ij}^{l}$, $\gamma_{ij}^{l}$, and $\delta_{ij}^{l}$ denote the spatial importance weights from four different levels of features. Similar to previous methods [27], we enforce $\alpha_{ij}^{l}+\beta_{ij}^{l}+\gamma_{ij}^{l}+\delta_{ij}^{l}=1$ and $\alpha_{ij}^{l},\beta_{ij}^{l},\gamma_{ij}^{l},\delta_{ij}^{l}\in \left[0,1\right]$. Therefore, for the spatial importance weight of $x_{ij}^{0\to l}$, we can define:(8)$$\alpha_{ij}^{l}=\frac{e^{\lambda_{\alpha_{ij}}^{l}}}{e^{\lambda_{\alpha_{ij}}^{l}}+e^{\lambda_{\beta_{ij}}^{l}}+e^{\lambda_{\gamma_{ij}}^{l}}+e^{\lambda_{\delta_{ij}}^{l}}}$$

Similarly, $\beta_{ij}^{l}$, $\gamma_{ij}^{l}$, and $\delta_{ij}^{l}$ are calculated through the softmax function, where $\lambda_{\alpha_{ij}}^{l}$, $\lambda_{\beta_{ij}}^{l}$, $\lambda_{\gamma_{ij}}^{l}$, and $\lambda_{\delta_{ij}}^{l}$ serve as the corresponding control parameters.

Unlike traditional methods that use fixed 1×1 convolutional layers to generate weight scalar maps, our ADFF method employs the dynamic convolution mechanism described in Section 3.2 to compute these weight scalar maps. Specifically, for each layer’s feature map $x^{n\to l}$, we utilize a dynamic convolution layer with *M* dynamic experts to generate the corresponding weight map $\lambda_{n}^{l}$. Compared to traditional convolution with fixed kernels, this dynamic convolution mechanism introduces only negligible computational overhead while significantly enhancing the model’s expressive capacity, thereby learning more optimal feature weight distributions.

## 4. Experimental Results

### 4.1. Experiment Settings

#### 4.1.1. Dataset

TT100K [7] is a traffic sign recognition dataset developed through collaboration between Tsinghua University and Tencent. The dataset was collected from real road scenes in Chinese cities and comprises 9176 high-definition images containing 30,000 traffic sign instances, with 6105 images allocated for training and 3071 for testing. Following the evaluation standards established by the COCO benchmark [48], we classify objects smaller than 32 × 32 pixels as “small objects”. This 0.15% threshold corresponds precisely to objects occupying less than 32 × 32 pixels in the TT100K dataset (as 32×32/(2048×2048)≈0.15% of the total image area). According to previous statistical analyses (Table 1) [8], more than 40% of traffic sign instances in TT100K belong to this small object category, presenting significant detection challenges due to multi-scale variations and complex background occlusions in traffic data. This characteristic distribution aligns exceptionally well with the primary research motivation of our study.

The dataset encompasses more than 200 types of traffic signs categorized by function and color into three major types: warning signs (yellow), prohibition signs (red), and indication signs (blue). A naming convention of “type+number” is adopted, such as “w34” representing warning sign number 34 “Beware of rear-end collision”, while “ph3.5” represents the prohibition sign “Height limit 3.5 m”. Despite the rich variety of classes, many categories have insufficient sample quantities; therefore, in practical research, we typically consider only the 45 classes that have at least 100 training samples each.

To further validate the generalization capability of our proposed method, we also conducted experiments on the CCTSDB2021 [49] dataset, which is an updated version of the Chinese Traffic Sign Detection Benchmark. This dataset contains traffic sign images collected from various road conditions across China, featuring three main categories: prohibitory signs, warning signs, and mandatory signs. CCTSDB2021 includes approximately 12,499 images in the training set and 5357 images in the validation set. The dataset presents diverse challenges including varying illumination conditions, weather effects, and different viewing angles, serving as a complementary benchmark to the TT100K dataset.

#### 4.1.2. Training Details

The experiments were conducted on the Ubuntu 22.04 operating system utilizing the PyTorch 2.5.1 deep learning framework. Our hardware configuration comprised one NVIDIA RTX 3090 GPU (NVIDIA Corporation, Santa Clara, CA, USA) and one Intel(R) Xeon(R) Gold 6430 CPU (Intel Corporation, Santa Clara, CA, USA). For the optimization strategy, this study employed the Stochastic Gradient Descent (SGD) optimizer for model training.

To determine the optimal hyperparameter configuration, we implemented a systematic parameter sensitivity analysis. As illustrated in Figure 8, we conducted a detailed evaluation of learning rates, testing three distinct values ranging from 0.001 to 0.1. Experimental results demonstrated that a learning rate of 0.01 yielded the optimal mean Average Precision (mAP) performance on the validation set. We configured the batch size to 32, which efficiently utilized the GPU memory resources (24 GB), achieving an actual memory consumption of 20.27 GB during operation. We established the optimal training duration at 200 epochs. As shown in Table 2, the model performance stabilized after approximately 200 epochs, with mAP50 reaching 0.9137. Extending the training beyond this point showed minimal improvements, with performance fluctuating within 0.07% through 300 epochs, indicating that 200 epochs is sufficient for model convergence.

#### 4.1.3. Evaluation Metrics

We evaluate our method using standard object detection metrics. Mean Average Precision (mAP) at an IoU threshold of 0.5 is used as the primary evaluation metric, representing the average AP across all classes. For efficiency assessment, we report Floating Point Operations (FLOPs) to measure computational complexity and Frames Per Second (FPS) to evaluate the real-time processing capability of the model. For real-time systems such as autonomous driving, a processing speed of at least 30 FPS is required to ensure that the model can analyze input data and respond in a timely manner [14].

### 4.2. Experimental Analysis

#### 4.2.1. Ablation Experiment of C3k2-Dynamic Module

To verify the effectiveness of the architectural choice of the C3k2-Dynamic module, we performed the following ablation study at a resolution of 640 × 640.

Table 3 demonstrates the impact of expert kernel quantity on performance. With M=2 expert kernels, the model shows limited expressivity. Increasing to M=4 substantially improves performance to 91.4% mAP. Further expansion to M=8 provides negligible benefits (only +0.1% mAP) while significantly increasing model complexity (+3.5 M parameters) and reducing inference speed (−8.9 FPS). This validates M=4 as the optimal balance point.

Table 4 compares different convolution kernel sizes. 1 × 1 convolutions are computationally efficient but have limited receptive fields; 3 × 3 convolutions achieve optimal performance while maintaining efficiency; 5 × 5 convolutions, despite larger receptive fields, result in decreased performance and slower processing speed. These experiments substantiate the rationality of selecting K=3.

Table 5 evaluates activation functions in the routing mechanism. ReLU, while computationally simple, tends to cause uneven weight distribution; Softmax ensures weight normalization but its mutual exclusivity constrains feature representation; Sigmoid enables multiple experts to activate simultaneously, achieving the highest detection accuracy (91.4%) while maintaining computational efficiency, confirming its suitability for the routing function.

#### 4.2.2. Analysis of Dilation Rate in Atrous Convolution

To validate the rationality of our selected dilation rate of 5 in FEM, we conducted a systematic ablation study at a resolution of 640 × 640 (Table 6) while keeping all other network parameters unchanged, examining the effects of various dilation rates.

As shown in Table 6, the performance initially improves with increasing dilation rate, reaching optimal results at a dilation rate of 5, which achieves 91.4% mAP50 and 71.9% mAP50-95. Further increases in dilation rate lead to performance degradation, likely due to excessive receptive field sizes capturing irrelevant contextual information.

#### 4.2.3. Ablation Study on Detection Head Configuration

To determine the optimal detection head configuration, we systematically compared the performance of different detection head combinations. In this study, the improved detection head specifically refers to our proposed ADFF detection head, while the conventional YOLO11 detection head serves as the baseline. As shown in Table 7, all network configurations incorporating the P2 detection head demonstrated improved mAP compared to the baseline model. We observed that merely enhancing the P3 to P5 detection heads without increasing the network’s detection resolution limited the maximum achievable mAP to 90.1%. Experimental results confirm that adding the P2 detection head effectively enhances the model’s capability to detect small objects, which is particularly evident in our improved architecture. After comprehensive evaluation of detection accuracy and real-time performance, we selected the improved {P2,P3,P4,P5} configuration as our final model architecture.

#### 4.2.4. Statistical Significance Analysis

To evaluate the stability and reproducibility of our proposed method, we conducted five independent training experiments with identical hyperparameters at a resolution of 640 × 640. The results, as presented in Table 8, demonstrate the high consistency of our approach, evidenced by the minimal standard deviations and narrow performance ranges across all metrics.

Furthermore, to comprehensively assess the model’s performance across various traffic sign categories, Table 9 presents the detailed mAP results of TSD-Net on 45 major classes within the TT100K dataset.

#### 4.2.5. Resolution Sensitivity Analysis

To evaluate the sensitivity of our proposed TSD-Net to input image resolution, we conducted systematic testing of model performance across different resolutions. We selected six distinct input resolutions—320 × 320, 416 × 416, 512 × 512, 608 × 608, 640 × 640, and 800 × 800—to analyze the impact of resolution variations on detection accuracy (mAP) and inference speed (FPS).

As shown in Table 10, with increasing input resolution, the model’s mAP exhibits a trend of rapid initial growth followed by gradual stabilization, while FPS demonstrates a significant downward trend. It is important to note that each incremental change (∆mAP and ∆FPS) is calculated relative to the previous resolution setting. Specifically, when resolution increases from 320 × 320 to 512 × 512, mAP improves substantially by 28.7%, whereas when resolution increases from 640 × 640 to 800 × 800, mAP increases by only 1.8% while FPS decreases dramatically by 22.1%.

Based on this analysis, we conclude that 640 × 640 represents the optimal working resolution for TSD-Net on the TT100K dataset, achieving a favorable balance between detection accuracy and inference speed.

#### 4.2.6. Ablation Experiments

To rigorously evaluate the performance enhancements introduced by our proposed methodology, we conducted a series of ablation studies. All experiments were performed using standardized 640 × 640 pixel input dimensions to ensure consistency across evaluations. The ablation studies were specifically conducted on the TT100K dataset, which provides a challenging benchmark for traffic sign detection with diverse scales and complex backgrounds.

Our experimental design systematically assessed the contributions of each proposed component: the C3k2-Dynamic module, FEM, improved FPN with the ADFF detection head. As illustrated in Table 11:

In Table 11, we collectively refer to the combination of the improved Feature Pyramid Network and ADFF detection head as iFPN and abbreviate the dynamic convolution C3k2 module as C3k2D for clarity of presentation. It is important to note that in the YOLO11s + FEM configuration, we retained the original feature pyramid structure of YOLO11, which means the FEM was added to the standard three-scale detection head architecture.

Based on the experimental data provided in Table 11, we conducted systematic ablation studies on the YOLO11s model using the TT100K dataset. We adopted the pre-trained weights of the YOLO11s model released by Ultralytics [38] as our baseline and conducted further training based on these pre-trained weights. Our choice of YOLO11s as the baseline was motivated by its excellent performance in general object detection tasks, its lightweight architecture suitable for deployment on edge devices in autonomous driving scenarios, and its open-source nature facilitating architectural modifications and performance comparisons. Our baseline model achieved 88.2% mAP50 at a frame rate of 128.2 FPS with a computational performance of 21.6 G FLOPs.

We evaluated each component individually for its independent contribution to the network performance: integrating the FEM module into the baseline model significantly increased mAP50 to 89.5%, albeit with increased computational complexity; independently introducing the iFPN module achieved 89.3% mAP50; while the incorporation of the C3k2D module successfully reduced FLOPs by 4.4% despite adding approximately 2.5 M parameters and moderately improved accuracy to 88.7%. The experimental results obtained provide substantial validation of the efficacy of the components proposed in this study in enhancing the performance of traffic sign detection. The final integrated model demonstrated a 3.2% improvement in mAP50 in comparison with the baseline. Although the inference speed decreased to 49.7 FPS, this performance still fully meets the real-time processing requirements for autonomous driving scenarios.

### 4.3. Comparative Experiments

To ensure fair comparison of speed and accuracy across all methods in the comparative experiments, we conducted inference time (FPS) measurements under identical hardware conditions. All experiments were executed on the same server equipped with an NVIDIA RTX 3090 GPU(NVIDIA Corporation, Santa Clara, CA, USA) and an Intel(R) Xeon(R) Gold 6430 CPU (Intel Corporation, Santa Clara, CA, USA).

#### 4.3.1. Comparison with Mainstream Traffic Sign Detectors

To verify the advantages of our proposed detection method, we compared it with the existing mainstream traffic sign detection methods on the TT100K dataset. To ensure fairness in the comparison, we excluded models utilizing pre-trained weights and detectors employing multi-scale input evaluation. As shown in Table 12, the previously best-performing AD-RCNN Lite [50] achieved an mAP of 86.3 but with a relatively low inference speed of 14.0 FPS. In contrast, our method attained an mAP of 91.4 at an input resolution of 640 × 640, demonstrating superior detection accuracy while maintaining real-time performance at 49.7 FPS. Although YOLOv4-tiny [51] achieved the highest frame rate of 89.0 FPS, its mAP was only 52.1, significantly lower than our method. This comparison demonstrates that our method achieves a better balance between accuracy and speed, making it more suitable for practical applications. A more intuitive comparison is shown in Figure 9.

To address the generalization concern, we further evaluated TSD-Net on the CCTSDB2021 dataset, which exhibits different characteristics from TT100K. The CCTSDB2021 dataset contains traffic sign images captured under various viewing angles (approximately ±30°), different distances (ranging from 5 to 50 m), and diverse weather conditions (including sunny, rainy, and cloudy). As shown in Table 13, under these varying conditions, TSD-Net demonstrated strong performance with an mAP of 99.8% at 640 × 640 resolution, suggesting its potential for generalization across different real-world scenarios.

#### 4.3.2. Comparison with Mainstream Object Detectors

We conducted a comprehensive evaluation of various object detection networks on the TT100K dataset, with detailed results presented in Table 14. To ensure evaluation fairness, we uniformly tested EfficientDet [60] and YOLO series networks using an input resolution of 608 × 608. For models such as M2Det [61], FPN [47], and RetinaNet [28], we maintained their original input dimensions. The experimental results demonstrate that among existing methods, YoloX-l achieved an mAP of 80.7 at 608 × 608 resolution, making it the best-performing model at this resolution. Our proposed method achieved an mAP of 90.8, significantly outperforming existing general-purpose object detectors. Notably, while our method’s inference speed (52.8 FPS) ranks in the middle range among mainstream object detection models, its detection accuracy substantially surpasses that of other general-purpose object detectors, as shown in Figure 10.

Similarly, we selected several methods for comparison with our model on the CCTSDB2021 dataset, as shown in Table 15. The experimental results demonstrate that our proposed method achieved an mAP of 98.5% on the CCTSDB2021 dataset, surpassing all other comparative models. Although the YoloV7 model exhibited impressive performance with a speed of 172.5 FPS and an mAP of 96.8%, our method still delivered a 1.7% improvement in detection accuracy. Compared to lightweight models such as YoloV5s and YoloX-s, our method provides superior detection precision while maintaining real-time performance capabilities.

In summary, our proposed TSD-Net based on YOLO11 not only surpasses the original model but also demonstrates superior performance compared to other mainstream object detection networks.

Figure 11 illustrates the detection results of our method on the TT100K dataset. Column a shows the original images, column b displays the detection results of YOLO11s, and column c presents the results of our proposed method. As evident from the figure, our method successfully identifies objects missed by YOLO11s, effectively resolving the false negative issue and improving detection accuracy for small-sized traffic signs. Our model successfully detects small-sized traffic signs that were previously undetectable due to YOLO11s’s resolution limitations (row 1) and addresses the challenge of incomplete recognition of adjacent small-sized traffic signs caused by insufficient resolution (row 2). Furthermore, our model demonstrates excellent performance in detecting traffic signs within complex backgrounds where YOLO11s fails to provide reliable recognition (row 3).

Given the numerous traffic sign categories in TT100K, we selected the ten traffic sign classes with the highest miss rates in YOLO11s for comparison with our model, as shown in Figure 12. We can clearly observe that TSD-Net consistently exhibits lower miss rates across various traffic sign categories compared to the YOLO11s model. Particularly noteworthy is the performance on w13 class signs, where TSD-Net achieves a miss rate of only approximately 0.10, representing a reduction of about 67% compared to YOLO11s’s 0.31. For pl60 and pl5 class signs, TSD-Net also maintains miss rates below 0.16 and 0.16, respectively, showing significant improvement over YOLO11s’s 0.26 and 0.25.

From an overall average performance perspective, TSD-Net achieves an average miss rate of approximately 0.15, whereas YOLO11s records 0.20, representing a reduction of about 25% in miss rate. These results convincingly demonstrate the superiority of our proposed TSD-Net architecture in handling traffic sign detection tasks, especially for small-sized traffic signs that are prone to being overlooked. It is worth noting that while TSD-Net significantly outperforms YOLO11s across most categories, the performance gap is relatively smaller for certain individual categories such as p3 and pl70. Nevertheless, considering the overall average performance, our improvement strategy still exhibits universality across most traffic sign types, effectively enhancing the reliability and robustness of the detection system.

### 4.4. Limitations Analysis

Despite the excellent performance of our proposed TSD-Net on the dataset, detection failures still occur in certain complex scenarios. To comprehensively evaluate the model’s limitations, we conducted a systematic analysis of failure cases across different scenarios, as shown in Figure 13.

We categorize the failure cases primarily into the following types:

Low-light Conditions. As shown in Figure 13a, under low-light conditions, the overall image brightness is significantly reduced, resulting in blurred color and texture features of traffic signs with diminished contrast, which impedes the model’s ability to accurately recognize them.

Severe Occlusion. As shown in Figure 13b, when traffic signs are partially occluded by trees, buildings, or other traffic signs, the integrity of their features is compromised, leading to a substantial decline in the model’s detection performance.

These failure case analyses provide important references for our future research directions. To address the aforementioned issues, we plan to further enhance the model’s robustness by introducing adaptive illumination enhancement modules, attention-guided occlusion handling mechanisms, and stronger contextual understanding capabilities.

## 5. Conclusions

This research presents the Traffic Sign Detection Network (TSD-Net), an innovative framework specifically designed to address key challenges in traffic sign detection. We incorporate three core components that significantly enhance small traffic sign detection performance: (1) The C3k2-Dynamic module, which adaptively adjusts convolution strategies based on input feature content, substantially improving the model’s adaptability to various traffic scenarios and sign types while expanding the model’s expression space without significantly increasing computational complexity. (2) The Feature Enhancement Module (FEM), which employs multi-branch convolution structures and dilated convolution techniques to expand the receptive field range, enhancing the model’s capability to capture target details. (3) The Adaptive Dynamic Feature Fusion (ADFF) detection head, which significantly improves small traffic sign detection through high-resolution detection branches and optimized cross-scale feature fusion mechanisms. Experiments on the TT100K benchmark dataset demonstrate that our method achieves 91.4% mAP50 at 640 × 640 resolution, surpassing the baseline YOLO11s model by 3.2 percentage points while maintaining real-time processing efficiency exceeding 49 FPS, meeting the practical requirements of autonomous driving scenarios. Compared to existing mainstream traffic sign detection methods, TSD-Net exhibits significant advantages in processing small-scale traffic signs under complex road backgrounds, providing robust support for the development of intelligent transportation systems. Future work will focus on further optimizing model robustness under extreme weather conditions and low-light scenarios while improving inference speed without sacrificing detection accuracy, thereby promoting the practical application and development of intelligent transportation systems.

## Figures and Tables

**Figure 1 sensors-25-03511-f001:**
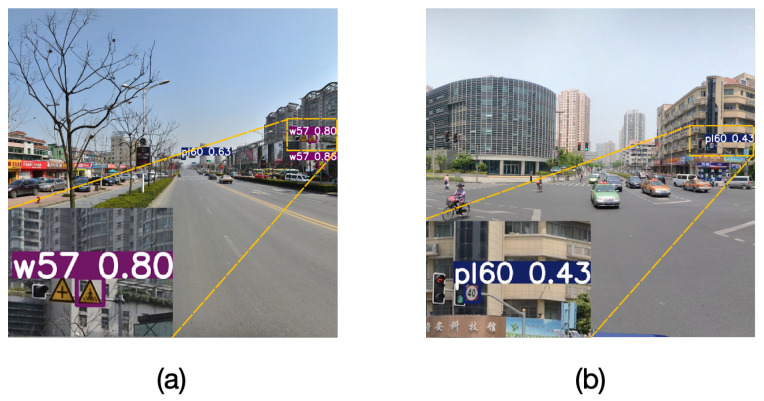
Challenges in traffic sign detection: (**a**) missed detections due to closely positioned small signs, (**b**) false positives due to complex background.

**Figure 2 sensors-25-03511-f002:**
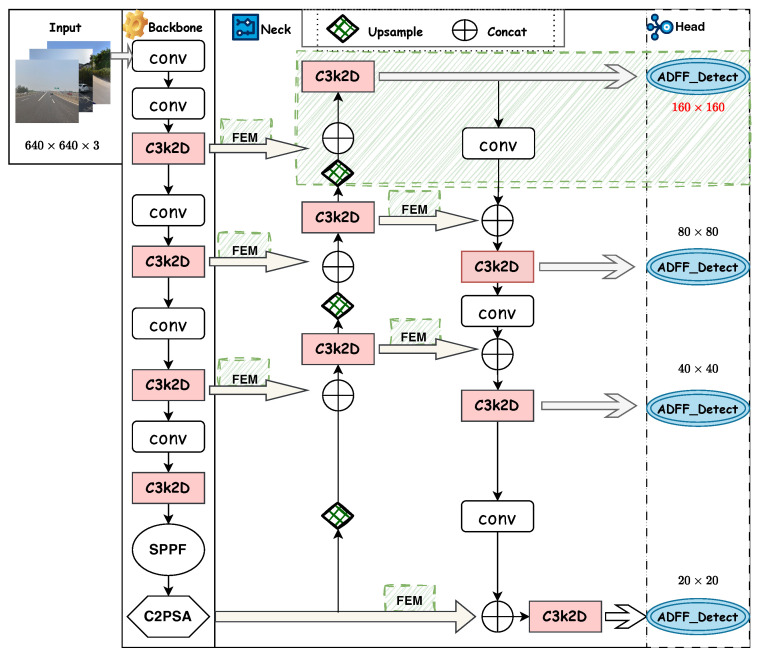
Network architecture of TSD-Net.

**Figure 3 sensors-25-03511-f003:**
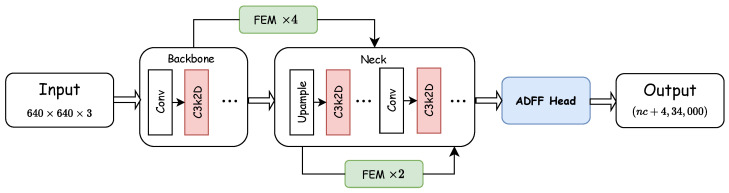
Component block diagram of TSD-Net.

**Figure 4 sensors-25-03511-f004:**
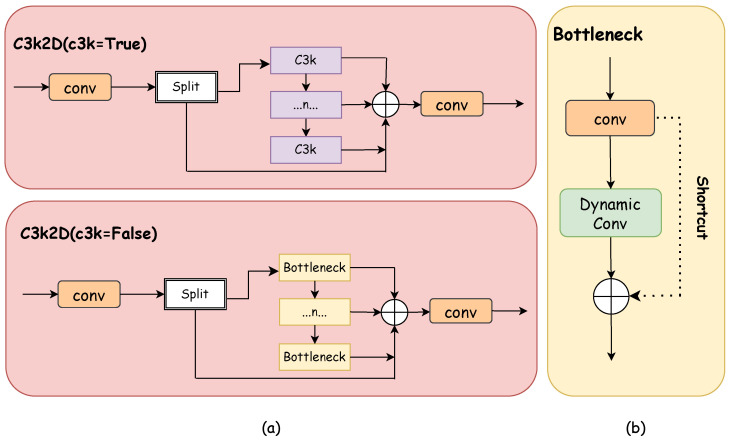
C3k2D design and internal structure. (**a**) C3k2D structure; (**b**) Dynamic bottleneck structure.

**Figure 5 sensors-25-03511-f005:**
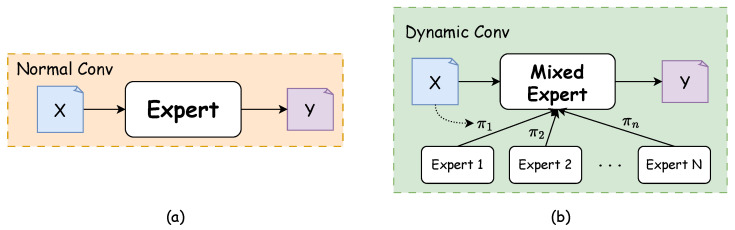
Comparison of different convolution structures. (**a**) Normal convolution. (**b**) Dynamic convolution (Adapted from Han et al. [25]).

**Figure 6 sensors-25-03511-f006:**
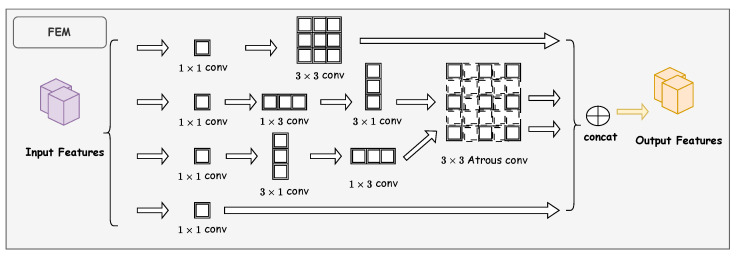
Structure of FEM (adapted from Zhang et al. [26]).

**Figure 7 sensors-25-03511-f007:**
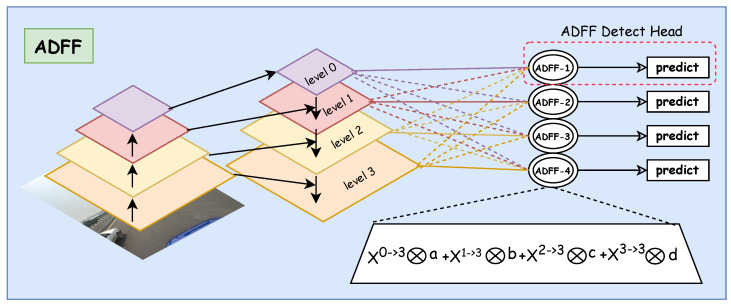
Illustration of the adaptive dynamic feature fusion mechanism.

**Figure 8 sensors-25-03511-f008:**
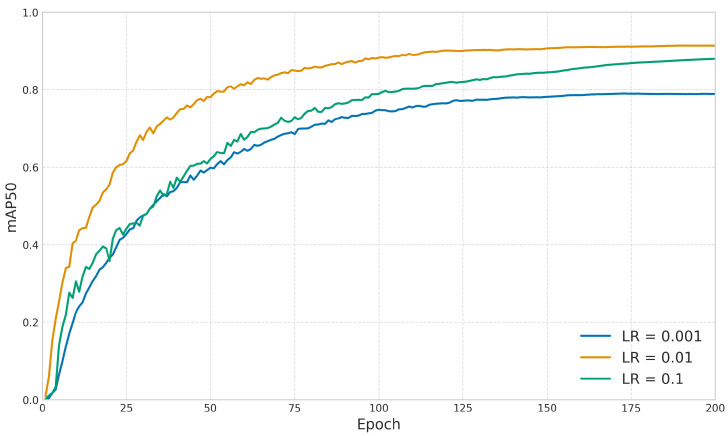
Training epoch sensitivity analysis for model convergence.

**Figure 9 sensors-25-03511-f009:**
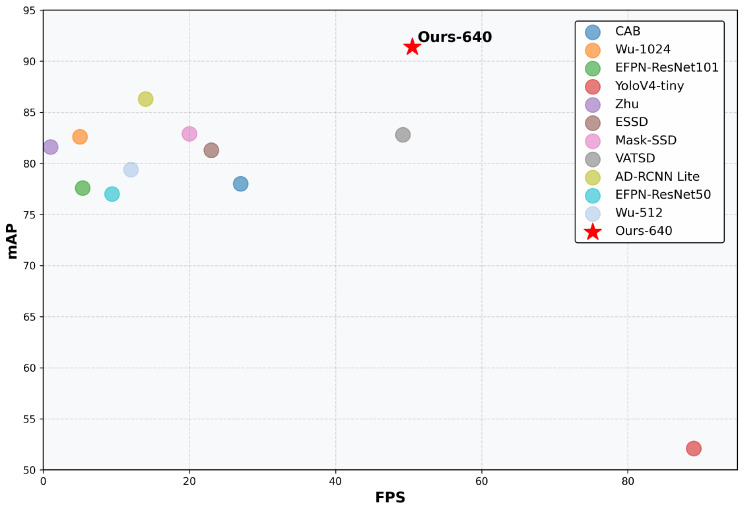
Performance comparison of different traffic sign detectors on TT100K dataset. The x-axis represents the inference speed (FPS) and the y-axis shows the detection accuracy (mAP).

**Figure 10 sensors-25-03511-f010:**
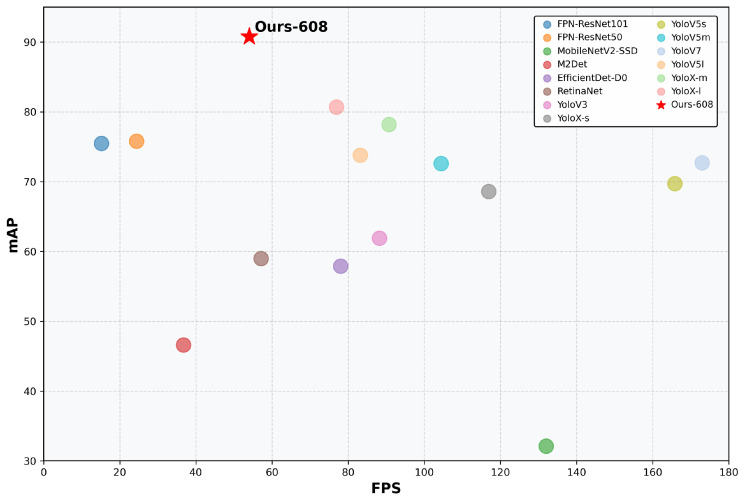
Performance comparison of different object detectors on TT100K dataset.

**Figure 11 sensors-25-03511-f011:**
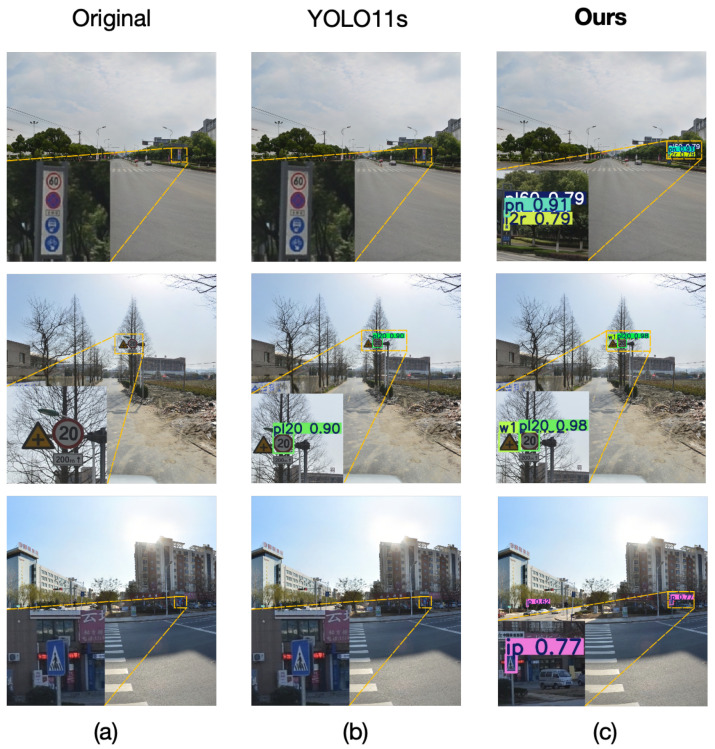
Traffic sign detection results on TT100K. (**a**) Original image; (**b**) Detection results using YOLO11s; (**c**) Detection results using our proposed method.

**Figure 12 sensors-25-03511-f012:**
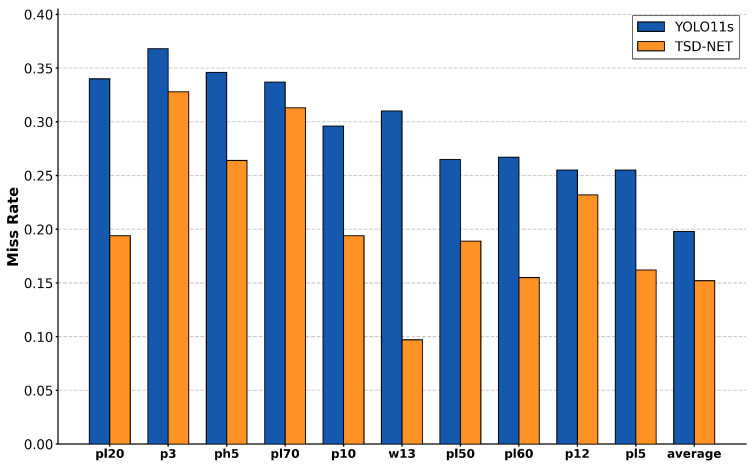
Comparison chart of miss rates in TT100k.

**Figure 13 sensors-25-03511-f013:**
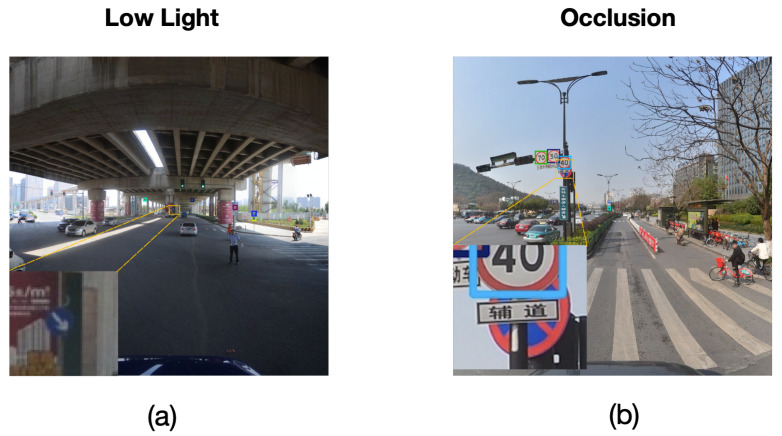
Typical failure cases of TSD-Net under different challenging scenarios: (**a**) low-light conditions and (**b**) severe occlusion.

**Table 1 sensors-25-03511-t001:** Traffic sign size distribution in the TT100K dataset.

Definition	Object Size	Instances	Proportion
Small	res <32×32	3281	43.5%
Medium and Large	32×32< res	4255	56.5%

**Table 2 sensors-25-03511-t002:** Training convergence analysis with different epoch settings.

Epochs	mAP50	Improvement	Epochs	mAP50	Improvement
50	0.7802	–	200	0.9137	+0.40%
100	0.8894	+13.99%	250	0.9131	−0.07%
150	0.9101	+2.32%	300	0.9136	+0.05%

**Table 3 sensors-25-03511-t003:** Ablation study on the number of expert kernels (M) in C3k2-Dynamic module.

Expert *M*	mAP50 (%)	FPS	Parameters
2	89.7	54.2	19.90 M
4	91.4	49.7	21.66 M
8	91.5	45.3	25.16 M

**Table 4 sensors-25-03511-t004:** Ablation study on kernel size in C3k2-Dynamic module.

Kernel Size	mAP50 (%)	FPS	Parameters
1 × 1	87.3	55.3	18.54 M
3 × 3	91.4	49.7	21.66 M
5 × 5	91.1	43.8	27.88 M

**Table 5 sensors-25-03511-t005:** Ablation study on activation functions in the routing mechanism of C3k2-Dynamic module.

Activation Function	mAP50 (%)	FPS
ReLU	89.7	49.8
Softmax	90.8	49.6
Sigmoid	91.4	49.7

**Table 6 sensors-25-03511-t006:** Ablation study on dilation rate in FEM.

Dilation Rate	Receptive Field	mAP50 (%)	${\mathrm{mAP}}_{50-95}$ (%)
1	3 × 3	88.7	68.3
3	7 × 7	90.6	70.8
5	11 × 11	91.4	71.9
7	15 × 15	90.5	70.5
9	19 × 19	89.9	69.8

**Table 7 sensors-25-03511-t007:** Number of detection heads and performance.

Detection Head	mAP (%)	Parameters	FPS
{P3,P4,P5}	89.5	14.60 M	89.5
{P2,P3,P4,P5}	90.5	16.85 M	76.3
Improved {P3,P4,P5}	90.1	19.20 M	62.1
Improved {P2,P3,P4,P5}	91.4	21.65 M	49.7

**Table 8 sensors-25-03511-t008:** Statistical analysis of TSD-Net performance.

Metric	Mean	Std Dev	Range (Min–Max)
mAP50 (%)	91.4	±0.15	91.2–91.6
mAP50–95 (%)	71.9	±0.16	71.8–72.1
FPS	49.7	±0.31	49.3–50.1

**Table 9 sensors-25-03511-t009:** Performance of TSD-Net on individual traffic sign categories in the TT100K dataset.

Class	mAP50	mA$P_{50-95}$	Class	mAP50	mA$P_{50-95}$	Class	mAP50	mA$P_{50-95}$
all	0.914	0.719	p11	0.949	0.698	pl40	0.915	0.710
pl80	0.930	0.768	i2r	0.916	0.690	i2	0.931	0.704
p6	0.935	0.723	p23	0.935	0.758	pl120	0.939	0.827
p5	0.950	0.775	pg	0.974	0.711	w32	0.815	0.538
pm55	0.894	0.743	il80	0.972	0.835	ph5	0.816	0.639
pl60	0.901	0.753	ph4	0.768	0.648	il60	0.988	0.838
ip	0.961	0.548	i4	0.977	0.687	w57	0.955	0.708
pl70	0.870	0.672	p13	0.910	0.652	pl100	0.960	0.836
pne	0.984	0.699	pr40	0.968	0.831	w59	0.810	0.621
ph4.5	0.922	0.782	pl20	0.879	0.683	il100	0.975	0.844
p12	0.903	0.752	pm30	0.731	0.628	p19	0.923	0.777
p3	0.828	0.716	pl30	0.905	0.736	pm20	0.852	0.711
pl5	0.908	0.668	pn	0.981	0.751	i5	0.969	0.708
w13	0.917	0.702	p26	0.930	0.749	p27	0.973	0.805
i4l	0.974	0.764	p10	0.840	0.647	pl50	0.917	0.680

**Table 10 sensors-25-03511-t010:** Resolution sensitivity analysis of TSD-Net.

Resolution	mAP (%)	FPS	ΔmAP	ΔFPS
320 × 320	68.6	82.7	–	–
512 × 512	88.3	65.8	+28.7%	−20.4%
608 × 608	90.8	52.8	+2.8%	−19.8%
640 × 640	91.4	49.7	+0.7%	−5.9%
800 × 800	93.0	38.7	+1.8%	−22.1%

**Table 11 sensors-25-03511-t011:** Model Ablation.

Modules				Performance			
**Baseline**	**FEM**	**iFPN**	**C3k2D**	**FLOPs (G)**	**Params (M)**	**mAP50 (%)**	**FPS**
✓				21.6	9.44	88.2	128.2
✓	✓			41.1	14.60	89.5	73.3
✓		✓		38.5	13.86	89.3	82.1
✓			✓	19.0	12.07	88.7	105.6
✓	✓	✓	✓	57.0	21.65	91.4	49.7

**Table 12 sensors-25-03511-t012:** Comparison of traffic sign detectors on the TT100k dataset.

Model	Resolution	mAP	FPS
CAB [52]	512 × 512	78.0	27.0
Wu et al. [53]	1024 × 1024	82.6	5.0
EFPN-ResNet101 [54]	1400 × 1400	77.6	5.4
YoloV4-tiny [51]	608 × 608	52.1	89.0
Zhu et al. [7]	2048 × 2048	81.6	1.0
ESSD [55]	512 × 512	81.3	23.0
VATSD [11]	608 × 608	82.8	49.2
Mask-SSD [56]	640 × 640	82.9	20.0
AD-RCNN Lite [50]	1024 × 1024	86.3	14.0
EFPN-ResNet50 [54]	1400 × 1400	77.0	9.4
Wu et al. [53]	512 × 512	79.4	12.0
YOLO-TSD [57]	-	81.2	-
RT-DETRmg [58]	-	83.1	-
DRC-YOLOv5s [59]	640 × 640	84.1	-
Our Method	608 × 608	90.8	52.8
Our Method	640 × 640	91.4	49.7

**Table 13 sensors-25-03511-t013:** Comparison of traffic sign detectors on the CCTSDB2021 dataset.

Model	Resolution	mAP	FPS
YoloV4-tiny [51]	608 × 608	89.7	88.0
ESSD [55]	512 × 512	90.5	23.7
VATSD [11]	608 × 608	93.2	49.2
Mask-SSD [56]	640 × 640	94.1	22.0
DRC-YOLOv5s [59]	640 × 640	96.7	-
Our Method	608 × 608	98.5	52.8
Our Method	640 × 640	99.8	49.7

**Table 14 sensors-25-03511-t014:** Comparison of object detectors models on the TT100k dataset.

Model	Resolution	mAP	FPS
FPN-ResNet101 [47]	1333 × 800	75.5	15.2
FPN-ResNet50 [47]	1333 × 800	75.8	24.4
MobileNetV2-SSD [62]	512 × 512	32.1	132.0
M2Det [61]	512 × 512	46.6	36.7
EfficientDet-D0 [60]	608 × 608	57.9	78.0
RetinaNet [28]	1000 × 800	59.0	57.1
YoloV3 [63]	608 × 608	61.9	88.2
YoloX-s [64]	608 × 608	68.6	116.9
YoloV5s [65]	608 × 608	69.7	165.8
YoloV5m [65]	608 × 608	72.6	104.4
YoloV7 [66]	608 × 608	72.7	173.0
YoloV5l [65]	608 × 608	73.8	83.2
YoloX-m [64]	608 × 608	78.2	90.7
YoloX-l [64]	608 × 608	80.7	76.9
Our Method	608 × 608	90.8	52.8

**Table 15 sensors-25-03511-t015:** Comparison of object detectors models on the CCTSDB2021 dataset.

Model	Resolution	mAP	FPS
RetinaNet [28]	1000 × 800	84.1	57.2
YoloV5s [65]	608 × 608	95.2	164.7
YoloV7 [66]	608 × 608	96.8	172.5
Faster R-CNN [4]	640 × 640	85.5	27.3
YoloX-s [64]	608 × 608	95.9	114.8
Our Method	608 × 608	98.5	52.8

## Data Availability

Data are contained within the article. The complete source code are available at https://github.com/Johnnymike20/TSD-Net, accessed on 20 May 2025.

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
