# Peer review of "TSD-Net: A Traffic Sign Detection Network Addressing Insufficient Perception Resolution and Complex Background"

_sensors, 2025, doi:10.3390/s25113511_

Round 1
Reviewer 1 Report
Comments and Suggestions for Authors
This manuscript introduces TSD-Net, a modified YOLO11-based object detector to enhance detection of small traffic signs in urban environments. The integration of C3k2-Dynamic modules, a Feature Enhancement Module (FEM), and an Adaptive Dynamic Feature Fusion (ADFF) detection head is well-motivated and demonstrated to improve performance on the TT100K dataset.
There are many useful parts of the paper:
A) The paper addresses a relevant and challenging problem in autonomous driving.
B) The architecture is technically novel and well-justified.
C) Empirical results show improved performance in both detection accuracy and speed.
D) Some ablation studies and comparisons strengthen the claims.
However there are a lot of areas that still need to fixed:
1. Training details are insufficient. The paper lacks clear reporting of data augmentation, weight initialization, learning rate schedules, and hyperparameter tuning strategies. These are essential for reproducibility. Alternatively, could the author make the code open to the readers?
2. Baseline model is not clear. The choice and description of the YOLO11 baseline are unclear. Is it an in-house variant or a known weights? This needs clarification and justification such baseline.
3. All experiments are conducted only on TT100K. It is unclear if and how TSD-Net would generalize to other traffic sign datasets, say taken at different resolution, angle of view, and/or distance.
4. The paper is overly verbose, with repetition and long explanations of standard concepts (e.g., AP, dynamic convolution). The paper could be more concise and focused on novel contributions.
5. While ablations are provided, it would be useful to test the sensitivity to image resolution and to include failure case analysis in more different settings and data types.
Overall, this paper has technical merit and good potential for publication but requires substantial revision to fix issues 1-5.
Reviewer 2 Report
Comments and Suggestions for Authors
This paper addresses the critical challenge of robust traffic sign detection in complex autonomous driving scenarios by proposing a novel TSD-Net model, which integrates C3k2-Dynamic modules, a Feature Enhancement Module (FEM), and an Adaptive Dynamic Feature Fusion (ADFF) head. While this topic is highly relevant to vehicle safety and fits well within the scope of the review, and the authors have proposed a comprehensive experimental design with promising reported improvements, major revisions are required to improve technical clarity and methodological justification. Here is a non-exhaustive list of recommendations:
1- The authors should clearly explain the novelty of this TSD-Net model compared to others used as traffic sign detectors in existing works, namely YOLOv8/11, TSD-YOLO, and transformer-based methods; while detailing how the C3k2-Dynamic module, the Feature Enhancement Module (FEM), and the Adaptive Dynamic Feature Fusion Head (ADFF) together overcome the limitations of previous work.
2- The authors should justify the choice of YOLO11 as a baseline and explain why pre-trained weights were not used, especially given their proven advantages related to convergence speed and final accuracy.
3- The authors should justify the choice of hyperparameters such as the initial learning rate (0.01), batch size (32), and the number of epochs (300). They should provide previous empirical evidence or ablation results demonstrating that these are the optimal parameters.
3- The authors should include the inference times (FPS) for all methods compared in Tables 2 and 3 on the same hardware to enable fair speed-to-accuracy comparisons.
4- The authors should clarify the rationale for defining "small objects" as occupying less than 0.15% (0.15% or 15% !) of the image area (i.e., why exactly 0.15%), and relate this threshold to the size distribution of TT100K PMC instances (instances range from 16 × 20 to 160 × 160 pixels).
5- The authors should report the mean and standard deviation of mAP and FPS over multiple training cycles to assess the statistical significance and reproducibility of the reported gains. Similarly, it is more accurate to include the mAP for each class and the mAP50-95.
6- The authors must justify the architectural choices of the C3k2-Dynamic module, including the number of expert kernels M, the kernel size, and the use of the sigmoid in the routing function, with a theoretical justification or an empirical comparison.
7- The authors must justify the dilation rate (5) used in the atrous convolution branches of the FEM by evaluating alternative rates or multi-rate schemes to demonstrate optimal receptive field design.
8- The authors must explain why a high-resolution P2 detection layer (4x downsampled, 160x160) was added rather than adapting the existing P3–P5 layers, and quantify its isolated contribution by ablation.
9- The authors should improve the clarity of the figures: ensure that Figures 2 to 6 include explicit module names, input/output dimensions, and annotated detection examples illustrating both typical successes and failure modes.
10- The authors should explain the computational load introduced by each component (FEM, iFPN, C3k2-Dynamic) by reporting the incremental changes in FLOP and FPS compared to the baseline YOLO11.
Reviewer 3 Report
Comments and Suggestions for Authors
Title: TSD-Net: A Traffic Sign Detection Network addressing insufficient perception resolution and complex background
This work proposes the Traffic Sign Detection Network (TSD-Net), a new framework designed to enhance the detection performance of small traffic signs in complex backgrounds. TSD-Net integrates a Feature Enhancement Module (FEM) to expand the network’s receptive field and improve its capability to capture target features. The work is good, and the authors have presented their work in detail. However, this manuscript currently needs some major corrections (describe below), that should be considered to be ready for publication.
Remarks to the Author: Please see the full comments.
1-Is the proposed work solved the following three limitations: high computational cost, inconsistent feature alignment, and insufficient resolution in detection heads? Please check and explain in a clear manner in the methodology section.
Also, it is recommended to replace the word “novel” with other synonyms.
2-The introduction section provides a good foundation on the topic of this research; however, it needs more improvement by adding more discussion about some recent works in this filed with a discussion of their gaps that lead to propose this work.
Besides, it is also preferable to avoid mentioning the proposed work in this section as that in Figure 1.
Moreover, ss stated on line 47, the phrase "Secondly,..." requires mentioning the first point. Please review such issues for the entire manuscript.
3- How was the "complex background interference" problem solved by adding the FEM module? Please provide more details. In fact, the entire "Materials and Methods" section should include a detailed explanation of each step to give a clear idea of ​​the proposed workflow.
4- A block diagram is required to illustrate the entire relationships between the various subsystems of the proposed work to simplify the reader’s understanding.
5- In fact, the paper contains some grammatical and linguistic errors, and these errors lead to a decrease in the reader's understanding, so the manuscript should be reviewed carefully.
6- Any information, figure, equation, or dataset taken from a previous source must be cited as a reliable source, unless it relates to the authors. Please, check this issue.
7- It is recommended to explain how the dynamic convolution mechanism is better than the traditional convolution with fixed kernels.
8- Can the authors provide further clarification and details about the data sets used in this work.
9- The proposed work is compared with other works, but it must be compared with other recent, relevant works to ensure its recent strength.
10- It is recommended that the conclusion be one paragraph and concise. In any scientific research, the conclusion should include the topics and data of the proposed work, summarize its main points, discuss its significance, and discuss future work. Please review all of these points to write a comprehensive conclusion.
Comments on the Quality of English Language
The English could be improved to more clearly express the research.
Round 2
Reviewer 2 Report
Comments and Suggestions for Authors
Clear and comprehensive answers to all recommendations.
Reviewer 3 Report
Comments and Suggestions for Authors
Most comments have been processed correctly. No further comments are needed.
Comments on the Quality of English LanguageThe English could be improved to more clearly express the research.